# Enhanced Ganoderic Acids Accumulation and Transcriptional Responses of Biosynthetic Genes in *Ganoderma lucidum* Fruiting Bodies by Elicitation Supplementation

**DOI:** 10.3390/ijms20112830

**Published:** 2019-06-10

**Authors:** Li Meng, Xiaoran Bai, Shaoyan Zhang, Mengfei Zhang, Sen Zhou, Irum Mukhtar, Li Wang, Zhuang Li, Wei Wang

**Affiliations:** 1Shandong Provincial Key Laboratory for Biology of Vegetable Diseases and Insect Pests, College of Plant Protection, Shandong Agricultural University, Tai’an 271018, China; mengli0121@126.com (L.M.); zs18562318828@163.com (S.Z.); 2Shandong Provincial Key Laboratory of Agricultural Microbiology, College of Plant Protection, Shandong Agricultural University, Tai’an 271018, China; 18769835516@163.com (X.B.); zhangshaoyan1995@163.com (S.Z.); ZMfei5566@163.com (M.Z.); 3Institute of Oceanography, Minjiang University, Fuzhou 350108, China; erumm21@yahoo.com

**Keywords:** medical fungi, ganoderma triterpenes, calcineurin signal, acetyl-CoA

## Abstract

Ganoderic acids (GAs) are a type of highly oxygenated lanostane-type triterpenoids that are responsible for the pharmacological activities of *Ganoderma lucidum*. They have been investigated for their biological activities, including antibacterial, antiviral, antitumor, anti-HIV-1, antioxidation, and cholesterol reduction functions. Inducer supplementation is viewed as a promising technology for the production of GAs. This study found that supplementation with sodium acetate (4 mM) significantly increased the GAs content of fruiting bodies by 28.63% compared to the control. In order to explore the mechanism of ganoderic acid accumulation, the transcriptional responses of key GAs biosynthetic genes, including the acetyl coenzyme A synthase gene, and the expression levels of genes involved in calcineurin signaling and acetyl-CoA content have been analyzed. The results showed that the expression of three key GAs biosynthetic genes (*hmgs*, *fps*, and *sqs*) were significantly up-regulated. Analysis indicated that the acetate ion increased the expression of genes related to acetic acid assimilation and increased GAs biosynthesis, thereby resulting in the accumulation of GAs. Further investigation of the expression levels of genes involved in calcineurin signaling revealed that Na^+^ supplementation and the consequent exchange of Na^+^/Ca^2+^ induced GAs biosynthesis. Overall, this study indicates a feasible new approach of utilizing sodium acetate elicitation for the enhanced production of valuable GAs content in *G. lucidum*, and also provided the primary mechanism of GAs accumulation.

## 1. Introduction

*Ganoderma lucidum*, also known as ‘the mushroom of immortality’ and ‘the symbol of traditional Chinese medicine’, is one of the best-known medicinal macrofungi in the world [1,2]. Ganoderic acids (GAs), which are C30 lanostane-type triterpenoids, are responsible for the pharmacological activities of *G. lucidum* [3]. Modern pharmacological research has demonstrated that GAs exhibit multiple therapeutic properties, including antitumour, antihypertensive, antiviral, and immunomodulatory activities [4,5,6].

GAs are synthesized via the mevalonate pathway (MVA), wherein acetyl-CoA is converted to 3-hydroxy-3-methylglutaryl-CoA (HMG-CoA) through a series of chemical reactions, and further to mevalonate (MVA) to isopentenyl pyrophosphate (IPP) to farnesyl diphosphate (FPP) to squalene, and finally to lanosterol [7,8,9]. The steps leading to the production of triterpenoids from lanosterol remain unknown, but probably include a series of oxidation, reduction, and acylation reactions. In pervious studies, 3-hydroxy-3-methylglutaryl coenzyme A reductase (HMGR) [10], farnesyl pyrophosphate synthase (FPS) [11], squalene synthase (SQS) [12], and oxidosqualene cyclase (OSC) [13] have been identified as key enzymes involved in triterpenoid biosynthesis, and their enhanced expression is found to promote the accumulation of GAs [14,15].

As the pharmacological activities of *Ganoderma* triterpenoids have been widely recognized, their sustainable production has become a necessity in order to fulfill the increasing demand. Various methods aimed at increasing the yields of GAs have been reported. For example, during the vegetative mycelia stage, considerable efforts have been made to improve GAs production in the submerged cultivation of *G. lucidum*, such as the optimization of cultivation conditions [16], the addition of elicitors [17,18,19], and the development of bioprocessing strategies [20,21]. Significantly, the key enzymes involved in triterpenoid biosynthesis have been observed to be upregulated during the treatment/induction process. Evidently, many factors influence GAs biosynthesis during the mycelia stage in *G. lucidum*. In general, the content of GAs in the fruiting body was significantly higher than that in the mycelia. However, it remains unclear whether the methods reported to increase GAs production in the mycelia stage could also be applied toward the fruiting body stage.

After the potential for acetate to be used as a fungal feedstock was discovered, efforts have been made to produce various biochemicals from acetate based on an understanding of its metabolism [22]. Extracellular acetate can be metabolized by microorganisms in the form of acetyl-CoA via two routes catalyzed by acetate kinase-phosphotransacetylase (AckA-Pta, encoded by *ackA* and *pta*, respectively) or acetyl-CoA synthetase (Acs, encoded by *acs*) [23]. Acetic acid is a substrate for acetyl-coenzyme A synthetase (acs) and acetyl-CoA, and is used in many metabolic pathways. Acetic acid can change the level of intracellular acetyl CoA through *acs* and then regulate intracellular metabolism and gene expression [24]. Channel ions were reported to have significant impacts on secondary metabolite biosynthesis in plants [25]. Sodium (Na^+^), the main channel ion, may have similar impacts on metabolite production. A high level of extracellular Na^+^ has been reported to increase intracellular Ca^2+^ to cope with salt stress in yeast [26] and rice [27] by triggering Ca^2+^ signals [28]. A similar stimulant effect by Na^+^ has been observed with respect to alatoxin [29] and GAs production in the mycelia phase [17]. In *Volvariella volvacea*, the mean weight increased when sodium acetate was applied as a spray during cultivation [30]. Therefore, it would be interesting to explore whether acetate improves GA production in fruiting bodies of *G. lucidum*.

The present study aimed to enhance GAs accumulation in the fruiting body of *G. lucidum* using sodium acetate as an inducer. To explore the underlying mechanisms responsible for its accumulation, the transcripts of genes encoding key enzymes in the GAs biosynthetic pathway, *acs* and those of genes in the calcineurin signal pathway, were also assayed.

## 2. Results

### 2.1. Effect of Sodium Acetate on the Dry Weight of Fruiting Body

The results showed that there was significant difference (*p* < 0.05) in the dry weights of the single fruiting bodies between the treatments and the control groups. The average dry weight of a single fruiting body was 8.96% to 16.64% higher than the control in the cases of sodium acetate and acetic acid treatments, respectively (Figure 1). However, no significant difference was observed in the shapes and diameters of the treated fruiting bodies (sprayed with sodium acetate or acetic acid) and control (Figure 2). These observations parallel those reported previously for *Volvariella volvacea* [30].

### 2.2. Effect of Sodium Acetate on Accumulation of GAs

The above results reveal that sodium acetate and acetic acid treatments increased the mean mushroom dry weight. To investigate whether GAs accumulation was also enhanced by sodium acetate or acetic acid treatments, GAs contents were measured in dried fruiting bodies. The GAs content increased from 26.93 to 34.64 mg/g in response to treatment with sodium acetate compared to control groups, representing an increase of 28.63% (*p* < 0.01) (Figure 3). In the case of acetic acid treatment, the accumulation of GAs in fruiting bodies was found to increase by only 8.44%, which indicated a non-significant difference from the GAs content in the control.

### 2.3. Effect of Sodium Acetate on the Expression of Key GAs Biosynthesis Genes

In Figure 4, it shows that in the sodium acetate-treated cultures at Day 40, the transcription levels of *hmgs*, *fps,* and *sqs* were higher than the control group by folds of 32.8, 6.9, and 12.0, respectively (*p* < 0.05). In the acetic acid treatment, the transcription level of *hmgs* was 4.9-fold higher (*p* < 0.05) than the control group. In treatment process, there were three key GA biosynthetic genes (*hmgs*, *fps,* and *sqs*) that were up-regulated under the influence of sodium acetate, while only one gene (*hmgs*) was up-regulated in the acetate acid condition. The results of qRT-PCR were consistent with the variation of GAs content in Figure 3. This study also demonstrates that acetic acid and sodium acetate influenced the mevalonate pathway in *G. lucidum*, leading to an increased accumulation of GAs.

### 2.4. Effect of Sodium Acetate on the Expression of acs and Acetyl-CoA Content

In order to survey the mechanisms in relation to enhanced GAs accumulation, we explored the expression of *acs* and measured the acetyl-CoA content in fruiting bodies of *G. lucidum* treated with sodium acetate or acetic acid. Under sodium acetate treatments, the transcription levels of homologous *acs* (*GL20510*, *GL21040,* and *GL24109*) were 6.96, 5.12, and 4.26-fold that of the control, respectively. In the acetic acid treatment, the transcription levels of homologous *acs* (*GL20510*, *GL23589,* and *GL24109*) were 4.32, 4.35, and 5.46-folds that of the control, respectively (Figure 5). In addition, the acetyl-CoA content was significantly higher in both treatments (Figure 6).

### 2.5. Effect of Sodium Acetate on the Expression of Genes in Calcineurin Signals Pathway

The process of Na^+^/Ca^2+^ exchange is important in cell physiology and metabolism. In order to survey whether this process influenced the accumulation of GAs, the expression levels of three sensor genes (*cam*, *cna,* and *crz1*) were quantified (Figure 7). The results illustrated that the transcription levels of *cna* and *crz1* were, respectively, 4.53 and 5.95-fold those of the control. However, the transcription levels of *cna* and *crz1* exhibited no significant responses to acetic acid treatment. Furthermore, the expression level of *cam* was down-regulated under acetic acid treatment. To clarify the relationship between media Na^+^ and calcineurin signals, the transcript levels of *ena1* and *Ca*^2+^*-ATPase* were measured in this study (Figure 7). The enzymes encoded by these two genes are responsible for the exchange of Na^+^ and Ca^2+^. In the sodium acetate treatment, the transcription levels of *ena1* and *Ca^2+^-ATPase* were, respectively, 1.89 and 3.32-fold that of the control. No significant differences in transcript levels were observed in the acetic acid treatment. Higher transcription levels of *ena1* and *Ca*^2+^*-ATPase* might facilitate ion homeostasis to respond to high Na^+^ stress.

## 3. Discussion

The bioactive compounds in medicinal mushrooms have been of great interest and investigated for a long time. In previously study, salicylic acid (SA) was reported to induce the biosynthesis of GAs in the mycelium of *G. lucidum* [7]. Recently, Liu et al. (2018) found that SA induces GAs biosynthesis by increasing reactive oxygen species (ROS) production, and further research found that *NADPH* oxidase-silenced strains exhibited a partial reduction in the response to SA, resulting in the induction of increased ROS production. Therefore, SA inhibits complex III activity to increase ROS levels and thereby induce the overproduction of GAs in *G. lucidum* [31]. GAs are an important microbial secondary metabolite that is largely produced in the fruiting body, but far less in the mycelium stage. Therefore, how to increase the accumulation of GAs in the fruiting body is worth studying in *G. lucidum*.

A recent study found that spraying SA during the fruiting body formation enhanced the accumulation of GAs. SA increased the triterpenoid content by 23.32% compared to the control [32]. In this study, sodium acetate was more effective (28.63%) at inducing the accumulation of GAs in the fruiting body of *G. lucidum*. However, the mechanism by which sodium acetate increased GAs biosynthesis largely remains unknown.

In order to survey the mechanisms by which sodium acetate increased the biosynthesis of GAs, we have explored the content of acetyl-CoA and gene expression levels in acetyl-CoA assimilation and the calcineurin signal pathway. The highest value of acetyl-CoA content, 398.52nmol/g, was recorded in the acetic acid treatment in Figure 6. This could be because acetic acid is a substrate for *acs*, and acetyl-CoA is used in many metabolic pathways [24]. However, the GA content was at its highest in the sodium acetate treatment, suggesting that acetyl-CoA may be only an initiator in the mevalonate pathway. The biosynthesis of GAs is also limited by the activity of other enzymes in the mevalonate pathway. In our study, the transcription levels of *hmgs*, *fps,* and *sqs* were highest in the sodium acetate treatment. Presumably, the higher transcript levels of GAs biosynthetic genes resulted in enhanced GAs accumulation, which is in agreement with what has been reported in other studies on the biosynthesis of GAs [9,16]. This may explain why the total GA content was highest in the sodium acetate treatment. Besides, our results suggest that increasing the acetyl-CoA content is not only way by which GAs accumulation could be manipulated. Therefore, we explored the effect of Na^+^ in inducing the biosynthesis of GAs. Here, we found that the transcription levels of genes encoding Ca^2+^/Na^+^ exchange and the genes in the calcineurin signal pathway were up-regulated in the sodium acetate condition. It indicates that while acetate ions directly enhance the biosynthesis of GAs, Na^+^ ions regulate the process of GAs biosynthesis.

In this study, the results suggested a hypothesis that both sodium and acetate ions participated in GAs biosynthesis. Acetate ions might alter the expression of genes related to *acs* and acetyl-CoA assimilation, thereby increasing GAs biosynthesis. On the other hand, Na^+^ induction would be expected to enhance cytosolic Ca^2+^ to induce GAs biosynthesis through the calcineurin signal pathway to up-regulate its biosynthetic genes at the transcriptional level (Figure 8).

## 4. Materials and Methods

### 4.1. Strains and Culture Conditions

*G. lucidum* strain (accession number: ACCC53264) was obtained from the Agricultural Culture Collection of China. It was grown at 28 °C in potato dextrose agar medium for 7 days. Spawn was prepared in polypropylene bags (18 cm in width × 36 cm in length). The growing medium was sawdust (78%), mixed with 20% wheat bran and 2% gypsum based on dry weight, and subsequently adjusted to a moisture content of 65%. The bags were filled with moistened substrate, sterilized at 121 °C for 2 h, and subsequently allowed to cool to room temperature. The inoculated substrate was incubated at 28 °C for mycelial colonization. The air temperature in the cultivation room was maintained at 28 °C and 80–85% relative humidity. The bags were sprayed with 20 mL of either sodium acetate (4 mM) or acetic acid (4 mM) at regular intervals of 4 days for a total period of 40 days. Control bags were sprayed with distilled water. When the white margins of the pileus disappeared, mature fruiting bodies were collected from the bags, weighed, and dried at 60 °C to a constant weight.

### 4.2. Measurement of Ganoderic Acids

Dried fruiting bodies were ground to a fine powder. Total GAs content were extracted from each sample separately by adding 1 g of grounded powder into 95% of ethanol (50 mL) at a frequency of 70 kHz in an ultrasonic bath for 2 h. After centrifugation at 4000× *g* for 10 min, one milliliter of the supernatant was obtained and mixed with 2 mL of 5% (*w*/*v*) vanillin and 5 mL of perchloric acid. The mixture was maintained in a water bath at 60 °C for 20 min, cooled in ice water for 10 min, and then followed by adding 200 μL of glacial acetic acid in 32 μL of the mixture. The total GAs content was determined based on a colorimetric reaction involving vanillin-perchloric acid and glacial acetic acid with GAs. The absorbance was measured at 550 nm with a spectrophotometer using ursolic acid as a reference.

### 4.3. Transcriptional Analysis

Total RNA was extracted from fruiting body samples of *G. lucidum* using RNAiso Plus (TaKaRa), and then treated with RNase-free DNaseI (TaKaRa). Synthesis of cDNA was with a TransStart All-in-One first-strand cDNA synthesis supermix for quantitative real-time PCR (qRT-PCR) according to the manufacturer’s protocol. The transcript levels of hydroxymethylglutaryl-CoA synthase (*hmgs*), hydroxymethylglutaryl-CoA reductase (*hmgr*), farnesyl pyrophosphate synthase (*fps*), squalene synthase (*sqs*), oxidosqualene cyclase (*osc*), and sensor genes (*cam*, *cna,* and *crz1*), encoding Na^+^-ATPase (*ena1*) and *Ca*^2+^*-ATPase*, were determined by qRT-PCR based on a relative method using a LightCycler^®^ 96 SW 1.1 with SYBR Green. Post-qRT-PCR calculations analyzing relative gene expression levels were performed according to the 2^−∆∆Ct^ method [33]. The gene expression level of 18S rRNA was found to be stable under experimental conditions; therefore, the transcription levels of different genes were analyzed by the standard-curve method and normalized by the *G. lucidum* 18S rRNA gene in control samples. For each gene, the expression level of the reference sample was set as 1.0, and the transcript level was measured as the increase in its mRNA level over that of the reference sample. The primer sets were used as previously described and are listed in Table 1. Genomic sequence alignment was used to identify 10 *acs* homolog genes (accession numbers in *G. lucidum* gene models: GL19651, GL20510, GL20899, GL21040, GL23180, GL23589, GL23735, GL24109, GL28494, and GL30345) in the *G. lucidum* genome [2].

### 4.4. Extraction and Measurement of Acetyl Coenzyme A (Acetyl-CoA)

Acetyl-CoA in fruiting bodies was extracted and measured using an acetyl-CoA assay kit (Solarbio, China). Briefly, fruiting body samples (100 mg) were ground in a mortar in an ice bath according to the manufacturer’s protocol, and then centrifuged at 8000× *g* for 10 min. Acetyl-CoA content was then measured at 340 nm, using a spectrophotometer as described by the protocol.

### 4.5. Experimental Design and Statistical Analysis

A completely randomized design with 30 replicates was used. All the data are expressed as the mean ± standard deviation (SD). Data were subjected to analysis of variance using the Student’s t-test, and the mean values indicating statistical significance were compared by Duncan’s multiple-range test using SPSS 17.0 software.

## 5. Conclusions

GAs are an important secondary metabolite that display mainly active content in *G. lucidum*. In the present study, we have investigated the inducing effect on GAs biosynthesis during fruiting body formation using sodium acetate. The results showed that sodium acetate not only enhances the GAs content, but also can increase the yield of fruiting body in *G. luncidum*. This study represents an important stimulus for future research on the biosynthesis and cultivation of GAs in the medicinal fungi *G. luncidum*.

## Figures and Tables

**Figure 1 ijms-20-02830-f001:**
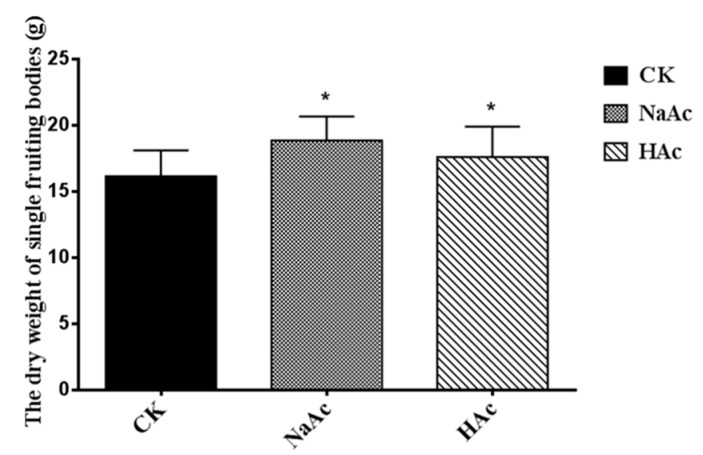
The average dry weights of single fruiting bodies. CK, control check; NaAc, treatment with sodium acetate; HAc, treatment with acetic acid. The error bars indicate the standard deviations from three independent samples. * indicates statistical significance (*p* < 0.05) compared to the CK strain.

**Figure 2 ijms-20-02830-f002:**
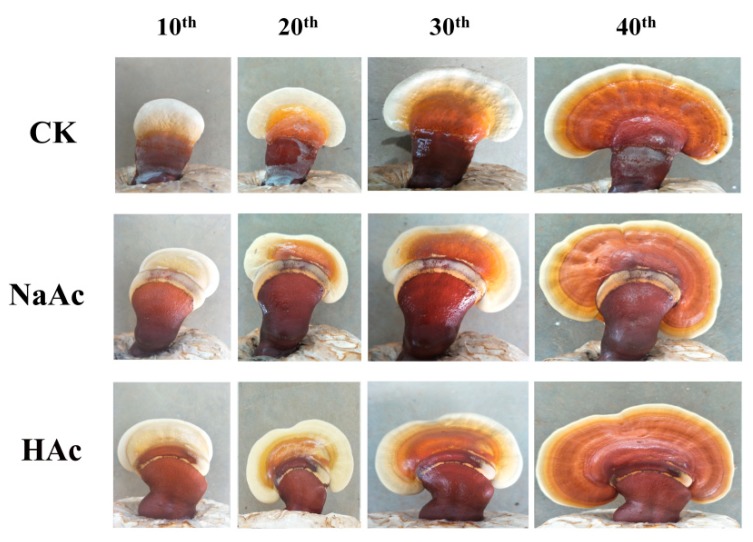
The morphology of fruiting bodies treated with sodium acetate and acetic acid in *Ganoderma lucidum*. CK, control check; NaAc, treatment with sodium acetate; HAc, treatment with acetic acid; the 10th, 20th, 30th, and 40th represent the 10th day, the 20th day, the 30th day, and the 40th day after treating.

**Figure 3 ijms-20-02830-f003:**
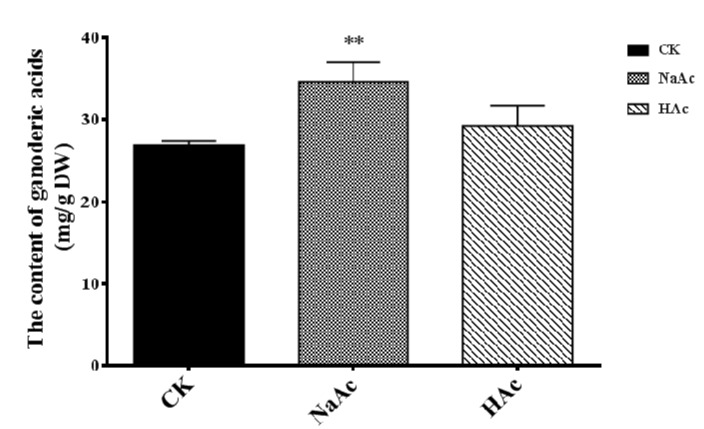
Accumulation of ganoderic acids under the influence of sodium acetate and acetic acid. CK, control check; NaAc, treatment with sodium acetate; HAc, treatment with acetic acid. The error bars indicate the standard deviations from three independent samples. ** indicates statistical significance (*p* < 0.01) compared to the CK strain.

**Figure 4 ijms-20-02830-f004:**
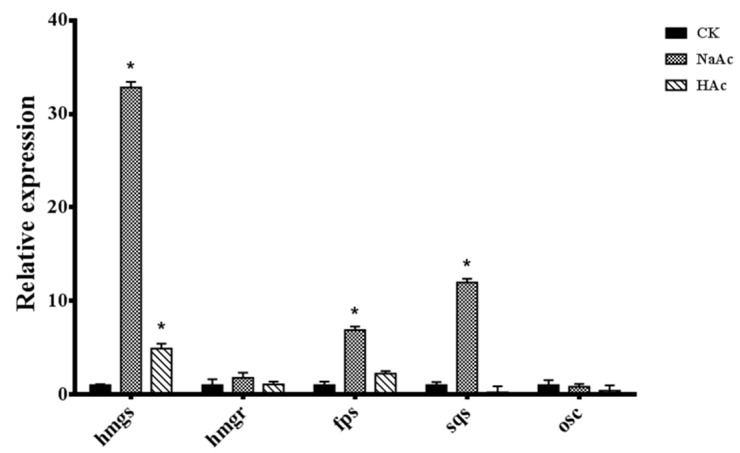
Transcriptional levels of the genes in the triterpenoid biosynthetic pathway. The expression level of the samples from the CK strain is defined as 1.0, and expression levels in the treatment strain are displayed in relation to the reference sample. The error bars indicate the standard deviations from three independent samples. * indicates a statistical significance (*p* < 0.05) compared to the CK strain. CK, control check; *hmgs*, hydroxymethylglutaryl-CoA synthase; *hmgr*, 3-hydroxy-3-methylglutaryl coenzyme A; *fps*, farnesyl pyrophosphate synthase; *osc*, oxidosqualene cyclase; *sqs*, squalene synthase; HAc, acetic acid; NaAc, sodium acetate.

**Figure 5 ijms-20-02830-f005:**
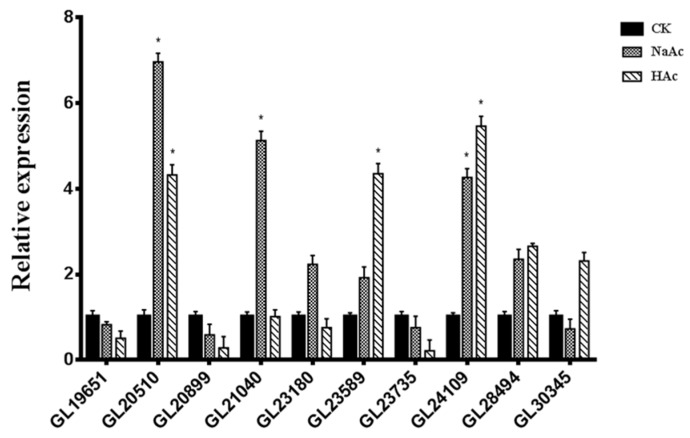
Transcriptional levels of the homologous *acs* genes in the CK strain and the treatment group. The expression level of the samples from the CK strain is defined as 1.0, and the expression levels in the treatment strain are displayed in relation to the reference sample. The error bars indicate the standard deviations from three independent samples. * indicates statistical significance (*p* < 0.05) compared to the CK strain.

**Figure 6 ijms-20-02830-f006:**
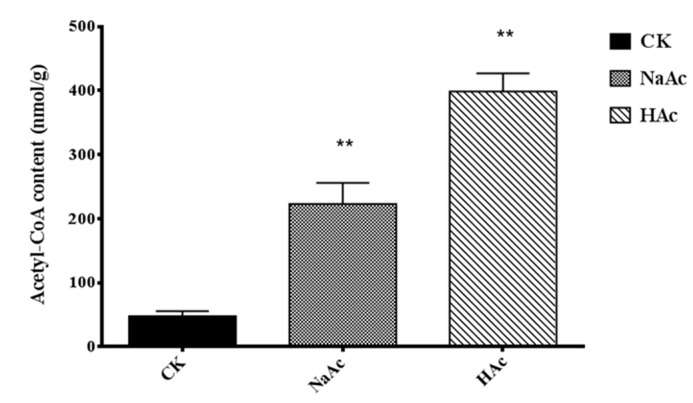
The acetyl-CoA content in mature fruiting bodies of *Ganoderma lucidum*. The error bars indicate the standard deviations from three independent samples. ** indicates statistical significance (*p* < 0.01) compared to the CK strain.

**Figure 7 ijms-20-02830-f007:**
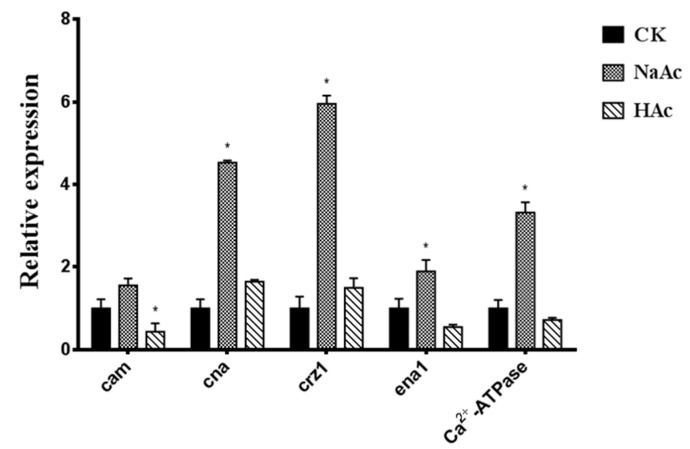
Transcriptional levels of calcineurin signal pathway genes in the CK strain and the treatment group. The expression level of the samples from the CK strain is defined as 1.0, and the expression levels in the treatment group strain are displayed in relation to the reference sample. The error bars indicate the standard deviations from three independent samples. * indicates statistical significance (*p* < 0.05) compared to the CK strain. CK, control check; HAc, acetic acid; NaAc, sodium acetate; *cam*, calmodulin; *cna*, calcineurin subunit A; *crz1*, calcineurin-responsive zinc finger; *ena1*, encoding Na^+^-ATPase.

**Figure 8 ijms-20-02830-f008:**
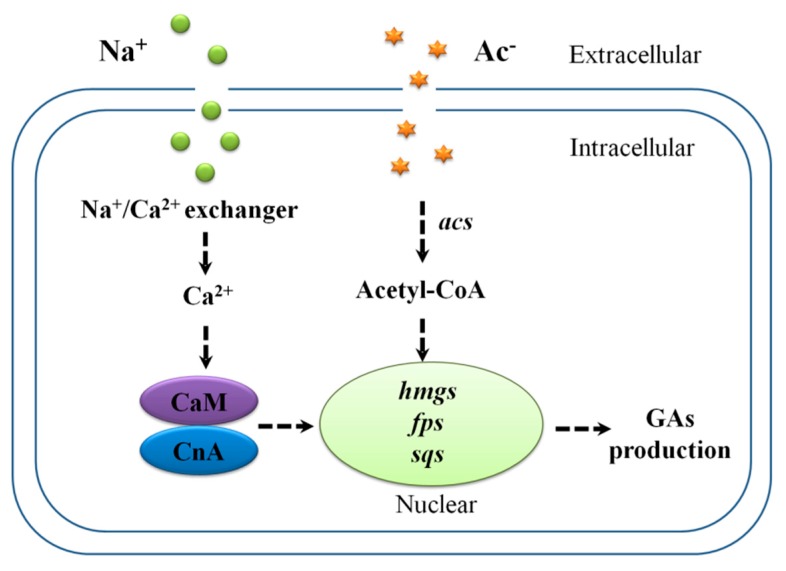
Schematic pathway predicting the role of sodium acetate-induced genes in *G. lucidum*. Dashed lines indicate steps consisting of multiple enzyme reactions. CaM, calmodulin; CnA, calcineurin subunit A; *acs*, acetyl-CoA synthetase; *hmgs*, hydroxymethylglutaryl-CoA synthase; *fps*, farnesyl pyrophosphate synthase; *sqs*, squalene synthase; GAs, ganoderic acid. Symbols: 
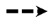
 increasing effect.

**Table 1 ijms-20-02830-t001:** Primer pairs used for qRT-PCR in this study.

Target Genes	Forward (5′–3′)	Reverse (5′–3′)	References
*18S rRNA*	TATCGAGTTCTGACTGGGTTGT	ATCCGTTGCTGAAAGTTGTAT	[17]
*hmgs*	CCCATCAACGCTTCCACCA	GCTCCTCCTCCGAAATGC	[9]
*hmgr*	GTCATCCTCCTATGCCAAAC	GGGCGTAGTCGTAGTCCTTC	[17]
*fps*	CCTCATCACCGCTCCAGAA	AGGGCGACGGGAAGGTAGAA	[9]
*osc*	AGGGAGAACCCGAAGCATT	CGTCCACAGCGTCGCATAAC	[17]
*sqs*	CTGCTTATTCTACCTGGTGCTACG	GGCTTCACGGCGAGTTTGT	[34]
*GL19651*	AGCAAGCAGTGGCATAA	GTCCCATCACGGTTCTC	[35]
*GL20510*	ATGGCGAAGGAAACCC	CGTCGGCCTCGTAGATG	[35]
*GL20899*	CCAGCACCACGAGTTAGG	GCGTTCGCCAGTCCAAA	[35]
*GL21040*	GCTAACATCGTCCAAGTCG	ATAGTCAACCCAGCAAACA	[35]
*GL23180*	CAGGAGTTCTTGTCGGTTGC	TCGTTCGTGGCGAGGTAG	[35]
*GL23589*	TCGCCTGCTATTCCATTC	ACGGCAACAGTCGTGAGT	[35]
*GL23735*	CGTCTGTTCGTCCACCC	GCGAGCGACTTTCCTGTT	[35]
*GL24109*	AGGCGGTTGATGTTCG	TCATGCCATACGCTACG	[35]
*GL28494*	GGCGTTTGTCATCCTCC	CCTTCTGAATCTTGCCTGTT	[35]
*GL30345*	AGTGACCGTTCGTGTTCC	GTAGGCTCCAGGTTCTCG	[35]
*ena1*	GACACGAAGACATCCTCACCC	CCATCCCATCCTCCCACTC	[17]
*Ca* ^2+^ *-ATPase*	GGCACTTATCCCCGTCCG	GATTGAGGGTCCGCCAGAG	[17]
*cam*	GAGGTACATCTCCGCCGCC	TCACGAACTCCTCGCAGTTGAT	[36]
*cna*	TCGGGGTCGTATAGGTCGG	CTTTGCGCTTTTGCGTGAG	[36]
*crz1*	GGGTGGCTGATGCAGAAATAC	CGAGGAGAGCGAGACGGG	[36]

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
