# Peer review of "Enhanced Ganoderic Acids Accumulation and Transcriptional Responses of Biosynthetic Genes in Ganoderma lucidum Fruiting Bodies by Elicitation Supplementation"

_ijms, 2019, doi:10.3390/ijms20112830_

Round 1

Reviewer 1 Report

The original scientific article titled "Enhanced ganoderic acid accumulation and transcriptional responses of biosynthetic genes in Ganoderma lucidum fruiting bodies by elicitation supplementation" is original and interesting to scienfific community. I recommend  acceptance in present form.

Line 227: much instead mush

Author Response

Response to Reviewer 1 Comments  

Point 1: Line 227: much instead mush 

Response 1: It has been revised. Much instead mush. (in red)

Reviewer 2 Report

Reviewer’s comments and suggestions for Authors

In this research manuscript, the authors have evaluated the method aimed at increasing the yields of Ganoderic acids (GAs). Previous studies have reported the significant findings that demonstrated GAs to be having multiple therapeutic activities, including antitumor, antihypertensive, antiviral and immunomodulatory activities. Therefore it has been intended to elevate the yields of GAs.

Basically, the study explores whether acetate improves GA production in fruiting bodies of G. lucidum. For this hypothesis, the authors have used sodium acetate as an inducer to enhance GA accumulation during fruiting body formation. Moreover, they have investigated the underlying mechanisms accountable for its accumulation and the transcripts of genes coding key enzymes in the GA biosynthetic pathway.

The study results suggested that three key GA biosynthetic genes (hmgs, fps, sqs) were significantly up-regulated. Analysis specified that the acetate ion increased the expression of genes related to acetic acid assimilation and increased GA biosynthesis, thereby resulting in GA accumulation. Further, the authors reported the expression levels of genes involved in calcineurin signaling.

The manuscript is well organized based on the study results. However, there few comments and advise to modify in the current version of the manuscript and submit again as a revised version.

1. In the abstract part of the paper, line number 26, there would be acetate ion.

2. In figure 4 and 7, the standard deviation was reported to be more. What would be the reason behind it? Please check if the authors can reduce the SD.

3. Line number 227 to 229, the authors need to rewrite the respective lines as the information was not clearly put up in the lines.

4. The figure needs to be redrawn as there positioning/ organization is not appropriate (as discussed in the study) both in figure and legends.

5. The first line of the conclusion needs to be modified as it is not appropriate in term of grammar.

6. The authors need to revise the manuscript by an English expert to reduce the grammatical errors. 

7. The authors need to check the references 2, 3, 5, 7, 15, 16, 34 as they were not following the journal guidelines (missing page number, some place used abbreviation (in journal name) others not why). 

Author Response

Response to Reviewer 2 Comments 

Point 1: In the abstract part of the paper, line number 26, there would be acetate ion. 

Response 1: It has been revised. Acetate ion instead acetat ion. (in red) 

Point 2: In figure 4 and 7, the standard deviation was reported to be more. What would be the reason behind it? Please check if the authors can reduce the SD. 

Response 2: It has been revised in figure 4 and 7.  The reason of the more standard deviation was because of three biological duplicates in qRT-PCR. In the revised manuscript, we recalculated the standard deviation using three technical duplicates. 

Point 3: Line number 227 to 229, the authors need to rewrite the respective lines as the information was not clearly put up in the lines. 

Response 3: It has been rewrite. (in red) 

Point 4: The figure needs to be redrawn as there positioning/ organization is not appropriate (as discussed in the study) both in figure and legends. 

Response 4: The figure 8 has been revised. The position and organization have been modified in figure 8. The legends have revised according the new figure. 

Point 5: The first line of the conclusion needs to be modified as it is not appropriate in term of grammar.

Response 5: It has been revised. “GA is an important secondary metabolite and mainly active content in G. lucidum.”(in red) 

Point 6: The authors need to revise the manuscript by an English expert to reduce the grammatical errors.

Response 6: We have revised the manuscript by an English expert Irum Mukhtar. (in red) 

Point 7: The authors need to check the references 2, 3, 5, 7, 15, 16, 34 as they were not following the journal guidelines (missing page number, some place used abbreviation (in journal name) others not why). 

Response 7: The references 3, 7, 15, 34 have been revised following the journal guidelines (the journal names were used abbreviation). However, the references 2 and 5 have only the paper number, and there are not having the page numbers. The reference16 was not found question. (in red)